# Bayesian Adversarial Learning

**Nanyang Ye**
University of Cambridge
Cambridge, United Kingdom
yn272@cam.ac.uk

**Zhanxing Zhu**$^*$
Center for Data Science, Peking University
Beijing Institute of Big Data Research (BIBDR)
Beijing, China
zhanxing.zhu@pku.edu.cn

## Abstract

Deep neural networks have been known to be vulnerable to adversarial attacks, raising lots of security concerns in the practical deployment. Popular defensive approaches can be formulated as a (distributionally) robust optimization problem, which minimizes a "point estimate" of worst-case loss derived from either per-datum perturbation or adversary data-generating distribution within certain pre-defined constraints. This point estimate ignores potential test adversaries that are beyond the pre-defined constraints. The model robustness might deteriorate sharply in the scenario of stronger test adversarial data. In this work, a novel robust training framework is proposed to alleviate this issue, Bayesian Robust Learning, in which a distribution is put on the adversarial data-generating distribution to account for the uncertainty of the adversarial data-generating process. The uncertainty directly helps to consider the potential adversaries that are stronger than the point estimate in the cases of distributionally robust optimization. The uncertainty of model parameters is also incorporated to accommodate the full Bayesian framework. We design a scalable Markov Chain Monte Carlo sampling strategy to obtain the posterior distribution over model parameters. Various experiments are conducted to verify the superiority of BAL over existing adversarial training methods. The code for BAL is available at https://tinyurl.com/ycxsaewr.

## 1 Introduction

Recently, the resurgence of deep neural networks has achieved various breakthroughs in different domains, including computer vision, natural language processing and game playing. Though with these successful applications, deep learning models were found to be vulnerable to maliciously manipulated and almost imperceptible perturbation (to human vision systems) over the original data, leading to wrong predictions over the manipulated examples [25, 8, 1]. The maliciously perturbed examples are known as adversarial examples. This issue has raised serious concerns about the robustness of deep neural networks, particularly when deployed in the security-sensitive scenarios.

Many different mechanisms have been proposed to defense the adversarial attacks. The most popular one is called adversarial training, in which the generated adversarial examples are fed to the neural networks as augmented data to enhance the robustness [8, 26, 10]. Alternatively, adversarial training can be posed as a robust optimization problem [2], where a min-max optimization problem is solved [18, 13]. The inner maximization problem aims to find the worst-case adversarial perturbation $\eta$ for the original data $\mathbf{x}$, i.e. maximizing the loss $\ell(\mathbf{x} + \eta; \boldsymbol{\theta})$, where $\boldsymbol{\theta}$ is the model parameters. Then the outer minimization solves the optimal model parameter $\boldsymbol{\theta}$ given the worst-case perturbation.

Another important line of research towards achieving robust models is formulated as a distributionally robust optimization problem. The basic idea is to minimize the worst-case loss with respect to

---

$^*$Corresponding author

perturbed data distribution around the data-generating distribution with certain probabilistic constraints. In [7], the authors proposed to constrain the perturbed distribution via first and second order moment uncertainty. Alternatively, Sinha et.al [22] constructed a neighborhood of perturbed distribution around the true data-generating distribution defined by Wasserstein metric; and then solve the worst-case loss inside this neighborhood through the benign duality of Wasserstein distance.

Moreover, some works [3, 9, 4] explored the adversarial machine learning from the game theoretical perspective, where a two-player (an adversary and a learner) min-max game is formulated and the Nash equilibrium is computed. This strategy was applied to traditional shallow machine learning models, such as Logistic regression and support vector machines. However, since the conditions of unique Nash equilibrium and computational tractability is quite restricted, it is not easy to be extended to train robust deep neural networks.

Therefore, most of existing robust learning approaches described above solve a "point estimate" of either worst-case per-datum perturbation or adversary data-generating distribution within pre-defined constraints, e.g. restricted by the maximum norm of perturbation [8, 26, 10] or inside some fixed Wasserstein ball [22]. One obvious limitation of this point estimate is the ignorance of potential test adversaries beyond the pre-defined norm or distributional constraints. The model robustness might deteriorate sharply in the scenario of stronger test adversarial data, as illustrated by the experimental results provided in Section 5.

In order to overcome this issue, we introduce a novel framework for robust learning, Bayesian Adversarial Learning (BAL), a fully Bayesian treatment over the adversarial training. In BAL, *a distribution is assigned to the adversarial data-generating distribution* to account for the uncertainty of the data-generating process. The uncertainty directly helps to consider the potential adversaries beyond the point estimate in the cases of distributionally robust optimization. We also incorporate the uncertainty of model parameters to accommodate the full Bayesian framework. Then the posterior distribution of model parameters can be obtained via combining iterative Gibbs sampling and scalable stochastic gradient sampling approaches.

We provide an instantiation for this general framework to defense adversarial examples. Some variants of the BAL framework for generating adversarial data, semi-supervised and unsupervised learning are discussed. Finally, we verify the effectiveness of our proposal in experiments.

## 2 Bayesian Adversarial Learning

We formulate our framework of Bayesian adversarial learning as follows. Firstly, assuming we have observed training data $\mathcal{D} = \{\mathbf{x}_i, y_i\}_{i=1}^{N}$ sampling from some unknown probability distribution with $\mathbf{x}, y$ representing the input and label, respectively, and denote $\mathbf{X} = \{\mathbf{x}_i\}_{i=1}^{N}$. Considering the scenario involving two players, a *data generator* and a *learner*. We establish a non-cooperative game with the two players: the role of the data generator is to modify the training data $\mathcal{D}$ into a perturbed one $\tilde{\mathcal{D}}$ to fool the learner's prediction, while the learner tries to predict best on the newly generated data $\tilde{\mathcal{D}}$.

We introduce a fully Bayesian treatment of both data generating process and learner's prediction. Concretely, we put an energy-based probability distribution over $\tilde{\mathcal{D}}$, i.e. *distribution over data distribution*, to quantify the uncertainty of data generating process.

$$p(\tilde{\mathcal{D}}|\boldsymbol{\theta}, \mathcal{D}) \propto \exp\{L(\tilde{\mathcal{D}}; \boldsymbol{\theta}) - \alpha c(\tilde{\mathcal{D}}, \mathcal{D})\}, \tag{1}$$

where the full energy term consists of two parts. The first one is the loss function $L(\tilde{\mathcal{D}}; \boldsymbol{\theta})$, induced by predicting on the perturbed data $\tilde{\mathcal{D}}$ given the learner's strategy $\boldsymbol{\theta}$, i.e. the parameter of the learning model $p(y|f(\mathbf{x}; \boldsymbol{\theta}))$. And the other part $c(\tilde{\mathcal{D}}, \mathcal{D})$ can be interpreted as the cost of modifying data $\mathcal{D}$ to $\tilde{\mathcal{D}}$, and $\alpha$ is a hyperparameter to balance the cost of data perturbation and the prediction loss inside the full energy term. The configuration of the loss function $L(\cdot)$ and cost term $c(\cdot, \cdot)$ will be specified later.

Given the data generator's perturbed data, the learner samples its model parameter $\boldsymbol{\theta}$ from the following energy-based conditional distribution,

$$p(\boldsymbol{\theta}|\tilde{\mathcal{D}}) \propto \exp\{-L(\tilde{\mathcal{D}}; \boldsymbol{\theta})\}p(\boldsymbol{\theta}|\beta), \tag{2}$$

where $p(\boldsymbol{\theta}|\beta)$ is the prior distribution over the model parameter $\boldsymbol{\theta}$ and $\beta$ is the hyperparamter.

Through the Bayesian adversarial learning, we aim at obtaining a robust posterior over the learner's parameter $\boldsymbol{\theta}$ given the observed data, $p(\boldsymbol{\theta}|\mathcal{D})$. This can be achieved via a standard Gibbs sampling procedure, i.e. iteratively implementing sampling according to Eq (1) and (2), for example, in $t$-th iteration,

$$\tilde{\mathcal{D}}^{(t)}|\boldsymbol{\theta}^{(t-1)}, \mathcal{D} \sim p(\tilde{\mathcal{D}}|\boldsymbol{\theta}^{(t-1)}, \mathcal{D}) \tag{3}$$

$$\boldsymbol{\theta}^{(t)}|\tilde{\mathcal{D}}^{(t)} \sim p(\boldsymbol{\theta}|\tilde{\mathcal{D}}^{(t)}). \tag{4}$$

After burn-in period, the collected sample $\{\boldsymbol{\theta}^{(T)}, \tilde{\mathcal{D}}^{(T)}\}$ follows the joint posterior $p(\boldsymbol{\theta}, \tilde{\mathcal{D}}|\mathcal{D})$. And therefore, $\boldsymbol{\theta}^{(T)}$ naturally follows $p(\boldsymbol{\theta}|\mathcal{D})$ of our interest since $p(\boldsymbol{\theta}|\mathcal{D}) = \int p(\boldsymbol{\theta}, \tilde{\mathcal{D}}|\mathcal{D})d\tilde{\mathcal{D}} = \int p(\boldsymbol{\theta}|\tilde{\mathcal{D}})p(\tilde{\mathcal{D}}|\mathcal{D})d\tilde{\mathcal{D}}$, which implicitly establishes an infinitely large ensemble of adversarial data by integrating over $\tilde{\mathcal{D}}$. This distinguishes from the existing robust training approaches solving a "point estimate" of either worst-case per-datum perturbation or adversary data-generating distribution.

During the test time, given the data point $\mathbf{x}_*$, we have the robust Bayesian prediction with respect to the posterior, which can be approximated by Monte Carlo samples from $p(\boldsymbol{\theta}|\mathcal{D})$:

$$p(y_*|\mathbf{x}_*, \mathcal{D}) = \int p(y_*|\mathbf{x}_*, \boldsymbol{\theta})p(\boldsymbol{\theta}|\mathcal{D})d\boldsymbol{\theta} \approx \frac{1}{T}\sum_{t=1}^{T} p(y_*|\mathbf{x}_*, \boldsymbol{\theta}^{(t)}), \quad \boldsymbol{\theta}^{(t)} \sim p(\boldsymbol{\theta}|\mathcal{D}). \tag{5}$$

To make the proposed BAL framework be practically applied, we have to specify some key ingredients, including how to generate the data $\tilde{\mathcal{D}}$ based on $\mathcal{D}$, the configuration of the cost function $c(\tilde{\mathcal{D}}, \mathcal{D})$, and the learner's model family $f(\mathbf{x}; \boldsymbol{\theta})$ and $p(y|f(\mathbf{x}; \boldsymbol{\theta}))$. We attribute these configurations of BAL to be problem dependent, relying on what kind of robustness we aim to achieve.

In the following, we propose a particular instantiation of BAL targeting on providing stronger robustness to adversarial examples and better generalization performance than existing approaches. Other types of configuration of BAL with different purposes of modeling will be discussed in Section 4.

## 3 A Practical Instantiation of Bayesian Adversarial Learning

Motivated by the generation of adversarial examples, we design that the data generator only modifies the input $\mathbf{X}$, with the labels remained unchanged. The most straightforward strategy is to perturb each training point $\mathbf{x}_i$ to $\tilde{\mathbf{x}}_i$ probabilistically,

$$p(\tilde{\mathcal{D}}|\boldsymbol{\theta}, \mathcal{D}) = p(\tilde{\mathbf{X}}|\boldsymbol{\theta}, \mathbf{X}) = \prod_{i=1}^{N} p(\tilde{\mathbf{x}}_i|\boldsymbol{\theta}, \mathbf{x}_i) \propto \exp\left\{\sum_{i=1}^{N} \ell(\tilde{\mathbf{x}}_i; \boldsymbol{\theta}) - \alpha \sum_{i=1}^{N} c(\tilde{\mathbf{x}}_i, \mathbf{x}_i)\right\}, \tag{6}$$

where the loss function $L(\cdot)$ and the cost are separable with respect to each adversarially generated data sample. The cost function can be the negative log likelihood given the learner's prediction model, $\ell(\tilde{\mathbf{x}}; \boldsymbol{\theta}) = -\log p(y|f(\tilde{\mathbf{x}}; \boldsymbol{\theta}))$; and $f(\tilde{\mathbf{x}}; \boldsymbol{\theta})$ could be a neural network or other types of prediction models; and the log likelihood $\log p(y|f(\tilde{\mathbf{x}}; \boldsymbol{\theta}))$ could correspond to cross entropy for classification or square loss for regression. We define the cost function as the most commonly used form by $L_2$ distance between $\tilde{\mathbf{x}}$ and $\mathbf{x}$, $c(\tilde{\mathbf{x}}, \mathbf{x}) = \|\tilde{\mathbf{x}} - \mathbf{x}\|_2^2$.

The key difference from the existing robust optimization methods [21, 18, 22, 13, 7] is that, instead of finding a single worst-case perturbed data set, a probability distribution is put over the generated data distribution to account for the underlying uncertainty of data generation. This could help the other player, the learner, to fully capture all the possible cases of generated data. Consequently, it will enhance the learner's robustness to adversarial examples during the test time, particularly when the test adversary is beyond the considered point-estimated worst-case data perturbation during the traditional robust optimization.

Given the generated data set $\tilde{\mathbf{X}}$, the learner provides its optimal response by sampling the following the conditional distribution,

$$p(\boldsymbol{\theta}|\tilde{\mathbf{X}}) \propto \exp\left\{-\sum_{i=1}^{N} \ell(\tilde{\mathbf{x}}_i; \boldsymbol{\theta})\right\} p(\boldsymbol{\theta}|\beta). \tag{7}$$

Similarly to Eq. (3) and (4), we can obtain the posterior samples of $p(\boldsymbol{\theta}|\mathcal{D})$ by iterative Gibbs sampling given in Eq. (6) and (7). Various gradient-based Markov Chain Monte Carlo (MCMC) methods can be adopted in each Gibbs iteration.

| Algorithm 1: Bayesian Adversarial Learning |
|---|

1: **Input:** $T$: the number of Gibbs iterations; $C$: the friction term for SGAdaLD; $\eta_1, \eta_2$: the step size; $\tau$: the exponential averaging window for SGAdaLD; $S_{\tilde{\mathbf{x}}}$ and $S_{\boldsymbol{\theta}}$: number of samples (or Markov chains) for representing conditional distribution over $\tilde{\mathbf{x}}$ and $\boldsymbol{\theta}$, respectively; $M$: the number of inner iterations.
2: **for** $t = 1 \ldots T$ **do**
3:    Randomly sample a mini-batch of observed data, $\{\mathbf{x}_s\}_{s=1}^{S_{\tilde{\mathbf{x}}}}$.
4:    **for** $s = 1 \ldots S_{\tilde{\mathbf{x}}}$ **do**
5:       Generate a standard Gaussian sample, $\mathbf{n} \sim \mathcal{N}(\mathbf{0}, \mathbf{I})$;
6:       Initialize current Markov chain with $\mathbf{x}_s$; and obtain $\tilde{\mathbf{x}}_s$ by running SGLD updates for $M$ iterations:

$$\tilde{\mathbf{x}}_s \leftarrow \tilde{\mathbf{x}}_s - \eta_1 \left( \sum_{s=1}^{S_{\boldsymbol{\theta}}} \frac{\partial (\log p(y|f(\tilde{\mathbf{x}}_s; \boldsymbol{\theta}_s^{(t)})) - \alpha c(\tilde{\mathbf{x}}_s, \mathbf{x}_s))}{\partial \tilde{\mathbf{x}}} \right) + \sqrt{2\eta_1}\mathbf{n} \qquad (8)$$

7:    **end for**
8:    **for** $s = 1 \ldots S_{\boldsymbol{\theta}}$ **do**
9:       Generate a standard Gaussian sample $\mathbf{n} \sim \mathcal{N}(\mathbf{0}, \mathbf{I})$;
10:       Update the sample $\boldsymbol{\theta}_s^{(t)}$ by running SGAdaLD updates for $M$ iterations:

$$\hat{\mathbf{V}}_{\boldsymbol{\theta}} \leftarrow (1 - \tau^{-1})\hat{\mathbf{V}}_{\boldsymbol{\theta}} + \tau^{-1} \left( \sum_{s=1}^{S_{\tilde{\mathbf{x}}}} \frac{\partial \log p(y|f(\tilde{\mathbf{x}}_s; \boldsymbol{\theta}_s^{(t)}))}{\partial \boldsymbol{\theta}} \right)^2$$

$$\boldsymbol{\theta}_s^{(t)} \leftarrow \boldsymbol{\theta}_s^{(t)} - \eta_2^2 \hat{\mathbf{V}}_{\boldsymbol{\theta}}^{-1/2} \left( \sum_{s=1}^{S_{\tilde{\mathbf{x}}}} \frac{\partial \log p(y|f(\tilde{\mathbf{x}}_s; \boldsymbol{\theta}_s^{(t)}))}{\partial \boldsymbol{\theta}} \right) + (2C\eta_2^3 \hat{\mathbf{V}}_{\boldsymbol{\theta}}^{-1} - \eta_2^4 \mathbf{I})\mathbf{n} \qquad (9)$$

11:    **end for**
12: **end for**
13: Collect $\{\boldsymbol{\theta}_s^{(T)}\}_{s=1}^{S_{\boldsymbol{\theta}}}$ as the posterior samples of $p(\boldsymbol{\theta}|\mathcal{D})$.

## 3.1   Scalable Sampling

In the scenario of large-scale data sets, sampling a full size of dataset from the distribution $p(\tilde{\mathbf{X}}|\boldsymbol{\theta}, \mathbf{X})$ is prohibitively expensive. Therefore, we resort to *stochastic approximation*, i.e. in each iteration only sampling a small mini-batch of adversarial data points $\{\tilde{\mathbf{x}}\}_{s=1}^{S_{\tilde{\mathbf{x}}}}$ with size $S_{\mathbf{x}} << N$, to approximately represent the original conditional distribution of the full adversarial data. We employ the stochastic gradient Langevin dynamics (SGLD, [27]) to achieve the scalable sampling for generating adversarial samples. And for sampling the posterior of model parameters, we find that the naive stochastic gradient Hamiltonian Monte Carlo (SGHMC, [6]) sampler cannot efficiently explore the target density due to the high correlation between parameters. To deal with this issue, we adopt the scale adapted version of SGHMC, stochastic gradient adaptive Hamiltonian Monte Carlo (SGAdaHMC) proposed in [23], where a diagonal pre-conditioning matrix is specified to accelerate the mixing of the sampler. For simplicity, we do not use the momentum in SGAdaHMC. We denote the simplified algorithm as stochastic gradient adaptive Langevin Dynamics (SGAdaLD). This leads to the following update procedure,

$$\boldsymbol{\theta} \leftarrow \boldsymbol{\theta} - \eta^2 \hat{\mathbf{V}}_{\boldsymbol{\theta}}^{-1/2} \mathbf{g}_{\boldsymbol{\theta}} + \mathcal{N}(0, 2C\eta^3 \hat{\mathbf{V}}_{\boldsymbol{\theta}}^{-1} - \eta^4 \mathbf{I}), \qquad (10)$$

where $\mathbf{g}_{\boldsymbol{\theta}}$ is the stochastic gradient of the system, $\eta$ represents the stepsize, $C$ is the friction coefficient and the diagonal pre-conditioning matrix can be updated using an exponential moving average, $\hat{\mathbf{V}}_{\boldsymbol{\theta}} \leftarrow (1 - \tau^{-1})\hat{\mathbf{V}}_{\boldsymbol{\theta}} + \tau^{-1}\mathbf{g}_{\boldsymbol{\theta}}^2$, and the hyperparameter $\tau$ can be also chosen automatically according to [23]. More details for SGAdaHMC could be referred to [23].

Besides its capability of dealing with large-scale data, SGLD and SGAdaLD also demonstrate the benefits of efficiently exploring the multi-modality of a distribution, which exactly matches our goals of utilizing the full distribution over the adversarial data and model parameters.

Incorporating SGLD and SGAdaLD into the Gibbs procedures (6) and (7), and assuming a uniform prior $p(\boldsymbol{\theta}|\beta)$, we summarize our BAL method for this particular instantiation in Algorithm 1. Note that in order to realize a more accurate representation over $p(\tilde{\mathbf{X}}|\boldsymbol{\theta}, \mathbf{X})$ and $p(\boldsymbol{\theta}|\tilde{\mathbf{X}})$, we maintain a number of Markov chains in parallel for samples $\tilde{\mathbf{x}}$ and $\boldsymbol{\theta}$. And the end point of each chain is used collectively for next iteration. Particularly, the starting points of the Markov chains for $\tilde{\mathbf{x}}$ are set as the observed data $\mathbf{x}$ to accelerate the mixing during the sampling process.

## 3.2 Connection to Other Defensive Methods

In this section, we elaborate on the characteristics of our proposed BAL framework through comparing it with other existing defensive mechanisms.

- BAL is a fully Bayesian approach for improving the robustness of neural network, implicitly forming an infinitely large ensemble by integrating over the model parameters. Practically, its inference procedure can be approximated by a collection of Monte Carlo samples, as described in Eq. (5), which is different from the adversarial training using ensemble models proposed in [26], where different model architectures are assembled. In BAL, only a single network is employed and different model parameter configurations are adopted according to the posterior distribution $p(\boldsymbol{\theta}|\mathcal{D})$. Obviously, BAL can also be extended to multiple model architectures to further promote robustness.

- Bayesian neural network (BNN, [17] also constructs a posterior distribution over model parameters, recently used a defensive method for adversarial attacks in [19]. BAL distinguishes from BNN by introducing conditional distribution of adversarial data given original training data. This crucial difference considers the adversary during training, yielding stronger robustness to adversarial attacks than BNN. In the experiments, we also compare the performance of BAL with BNN to show the benefits of BAL.

- In parallel, the Bayesian GAN with a similar min-max formulation is proposed for generative modeling [20], not for adversarial learning. Different from stochastic gradient Hamiltonian Monte Carlo (SGHMC) method used in Bayesian GAN, which is found to be quite slow, we use the adaptive sampling method instead.

# 4 Variants of Bayesian Adversarial Learning

**Alternatives of Generating Adversarial Data**     Since BAL is a general framework designed for robust learning, besides the method of generating adversarial data shown in Eq. (6), other alternatives could also be implemented according to different purposes of robust modeling. For instance, instead of considering perturbing the data in a pixel-wise manner, we can generate the adversarial data by certain spatial transformation, as described in [29]. And the cost function $c(\tilde{\mathbf{x}}, \mathbf{x})$ can be specified accordingly. Alternatively, generative adversarial networks can also be designed for synthesizing adversarial data, such as [28]. We leave the exploration on the other types of generating adversarial data as future work.

**Robust unsupervised and semi-supervised learning via BAL**     There also exits adversarial attacks for unsupervised and semi-supervised models: in [14], the authors proposed several methods for attacking variational auto-encoder (VAE, [12]) and VAE-GAN [16]. Given this scenario, our BAL framework can also be adopted to achieve robust supervised and semi-supervised models through configuring an appropriate data generation distribution $p(\tilde{\mathbf{x}}|\mathbf{x})$.

Take training a robust VAE as an example, we can specify the data generator by sampling from the following distribution,

$$p(\tilde{\mathbf{x}}|\mathbf{x}) \propto \exp\left\{\ell(\mathbf{z}, f_{enc}(\tilde{\mathbf{x}}; \boldsymbol{\theta})) - \alpha c(\tilde{\mathbf{x}}, \mathbf{x})\right\}, \tag{11}$$

where $f_{enc}(\cdot)$ is the encoder of VAE parameterized by $\boldsymbol{\theta}$, and $\mathbf{z}$ is the latent representation of $\mathbf{x}$. Concretely, the data generator aims to synthesize a perturbed sample $\tilde{\mathbf{x}}$ such that its latent representation $f_{enc}(\tilde{\mathbf{x}}; \boldsymbol{\theta})$ deviates from $\mathbf{z}$ as much as possible. Then the parameter $\boldsymbol{\theta}$ of the encoder can be learned by sampling from $p(\boldsymbol{\theta}|\mathbf{X}) \propto \exp\left\{-L_{VAE}(\mathbf{X}; \boldsymbol{\theta})\right\} p(\boldsymbol{\theta}|\beta)$, where $L_{VAE}(\mathbf{X}; \boldsymbol{\theta})$ is the original VAE loss.

# 5 Experiments

To evaluate the proposed method, we conduct experiments on MNIST classification and traffic sign recognition. The comparing methods include standard empirical risk minimization (**ERM**), adversarial training with fast gradient sign method (**FGSM**, [8]), iterative FGSM (**IFGSM**, [15]) and the projected gradient method (**PGD**, [18]), **WRM** [22], Bayesian Neural Network (**BNN**) and our proposed **BAL**. The Monte Carlo samples for BNN is also based on SGAdaLD, and the same number of samples are collected to compare with BAL.

In the following experiments, for BAL, we set $S_{\tilde{\mathbf{x}}}$ to be 1 and $S_{\boldsymbol{\theta}}$ to be 5 for the mnist classification and $S_{\tilde{\mathbf{x}}}$ to be 5 and $S_{\boldsymbol{\theta}}$ to be 5 for the traffic sign recognition. For other adversarial training methods and ERM, to fully explore the parameter space, we use the Adam optimizer for minimizing the loss on adversarial examples generated by FGSM, IFGSM, PGD and WRM. We refer to the authors' code for reimplementing WRM [2]. The number of steps for IFGSM and WRM to generate adversarial examples is set to be the same as $M = 15$ in our algorithm for fair comparison. During adversarial training, we use the parameters of the well-trained ERM model for initialization, and train the model for extra two or one epoch. The reason for this efficient training strategy is that we found the results obtained from training with a large number epochs from scratch, are similar with that of training a small number of epochs initialized with ERM. We fixed the sampling temperature for SGLD to be $10^{-5}$ in the experiments. We use the sampling temperature of $10^{-5}$ for SGAdaLD in MNIST classification task and $10^{-7}$ for SGAdaLD in traffic sign recognition task.

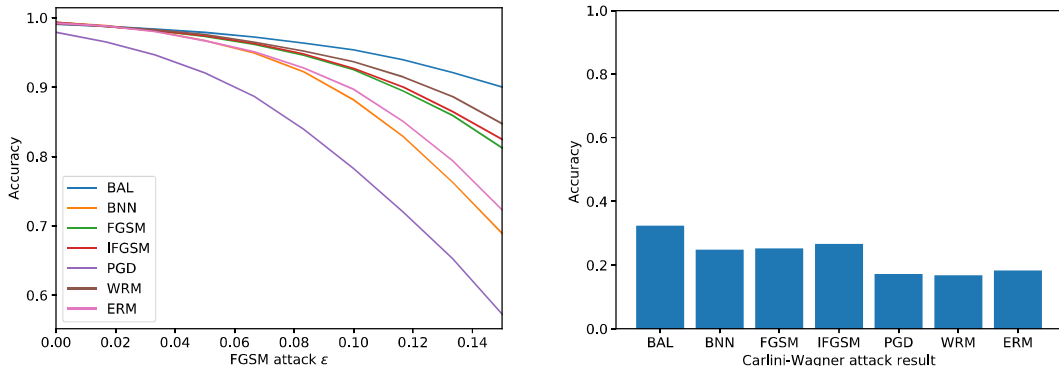

Figure 1: **Left:** Test accuracy on white-box attacks generated by FGSM method for MNIST classification; **Right:** Test accuracy on white-box attacks generated by Carlini-Wagner method for MNIST classification

## 5.1 MNIST Classification

We consider the standard MNIST classification. We use a neural network consisting of 32 3x3 convolutional filters, a 2x2 max-pooling layer, 32 3x3 convolutional filters, a 2x2 maxpooling layer, a dropout layer, a ReLU fully connected layers, a fully connected layer. Note that we use ReLU-based networks instead of ELU in [22] as they have achieved the state-of-the-art performance on many tasks.

We use FGSM to attack the adversarially trained models with different perturbation magnitude $\epsilon_{adv}$. The FGSM attack $\epsilon$ is set to be in the range from 0 to 0.15. The result is shown in Figure 1(a), from which we can observe that even with large perturbations on the input, our proposed method could still achieve good performance. Note that with perturbations to $\epsilon$ is smaller than 0.02, FGSM and IFGSM achieved the better performance than BAL as they are trained on a similar level. However, when $\epsilon$ becomes larger, BAL and WRM can achieve better performances than other methods, which is consistent with the original WRM's paper's results on MNIST dataset. Besides, the accuracy of BNN is close to the ERM model. This indicates that adding the Bayesian adversarial learning is necessary for robustness.

To further evaluate the BAL, we run a much stronger Carlini-Wagner L2 attack [5] on adversarially trained models to avoid over-fitting the FGSM adversarial examples. We found that the original CW implementation is too slow for evaluating on the whole test dataset, and the adversarial accuracy remains similar even when we tried to use longer iterations. As there are lots of hyper-parameters to tune, we decided to re-implement a simple one-step Carlini-Wagner L2 attack method instead. In our implementation, we use the SGD optimizer instead of the Adam optimizer to maximize the prediction probability difference of other classes and the correct class with a two-norm regularization term. This implementation is faster and can have stronger attack effects in our initial experiments with fewer steps. Figure 1(b) presents the prediction accuracy on the adversarial exampled generated by the Carlini-Wagner attack.

To analyze the effects of sample size $S_{\theta}$, we run the experiments sample size $S_{\theta}$ 1,3,5,10 and plot the results in Figure 2.

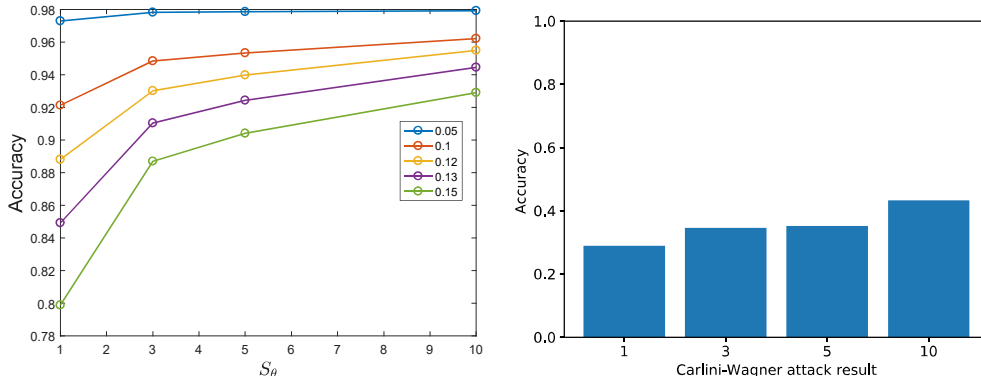

Figure 2: **Left:** Sample size test against FGSM attack; **Right:** Sample size test against Carlini-Wagner method.

From Figure 2, we can observe that under the same strength of attack, the accuracy grows with the increasing sample size as expected. We can also find that larger sample size can provide more protection against adversarial attack because larger sample size can represent the target posterior space better. Besides, increasing sample size can also improve adversarial defense performance even against the Carlini-Wagner method. We also tried different $S_z$ but did not find any systematic improvement with the increase of $S_z$ in this problem. We speculate that this is because that for MNIST dataset, a small number of samples could already represent the distribution well.

We use the same neural network architecture and run our experiments on the Fashion-MNIST dataset with a sample size of 5. The result is shown in Appendix A.

## 5.2 Spatial Transformation Network for Traffic Sign Recognition

To evaluate our method on more realistic safety-critical problems, we consider the robust learning for traffic sign recognition based on spatial transform neural networks (STN, [11]). Spatial transform networks could learn a geometrical transformation to make the model more robust to changes of orientation, translation. We use the STN combined with convolutional neural networks to classify traffic signs from the gray transformed German Traffic Sign Benchmarks (GTSRB, [24]) with 43 classes.

To provide a fair comparison between different methods, we tune the hyperparameters of each method such that they can achieve around 94% test accuracy on the clean test data except for BAL to be around 97%. In the experiments, we run FGSM white-box attack with $\epsilon$ ranging from 0 to 0.1 as we found that when the FGSM attack $\epsilon$ is above 0.1, adversarial samples will be indistinguishable even for human beings. The result is shown in Figure 3(a). From this result, we can conclude that our proposed BAL can have better performance in adversarial training in more complex and realistic scenario. When $\epsilon$ is large, BAL can still achieve good prediction accuracy compared with other methods. Note that PGD's performance is not good in this setting since it is required to achieve high accuracy on clean data. To make adversarial training methods useful in realistic settings, the clean accuracy should be high enough. However, with this constraint, we find that it is very hard for

adversarial training with PGD to have strong adversarial robustness. The closeness of other methods' performance indicates that when we set a strict constraint on the clean data accuracy, simply using larger adversarial perturbations cannot have performance guarantee on adversarial examples while BAL can provide an alternative way for adversarial defense within the BAL framework. We further evaluate all methods' robustness against the simplified one-step Carlini-Wagner attack as the previous experiment. We use the same setting as the previous experiment. The results are shown in Figure 3(b). From Figure 3(b), we can conclude that the all methods' accuracies have been reduced to a low level as the Carlini-Wagner attack is very strong. However, under this setting, our method still reaches an accuracy of 7.41%, much higher than the random guess accuracy of 2.3%. This is very important for safety-critical systems in real practice as even under the worst scenario, we can still rely on the BAL trained system to provide predictions better than the random guess.

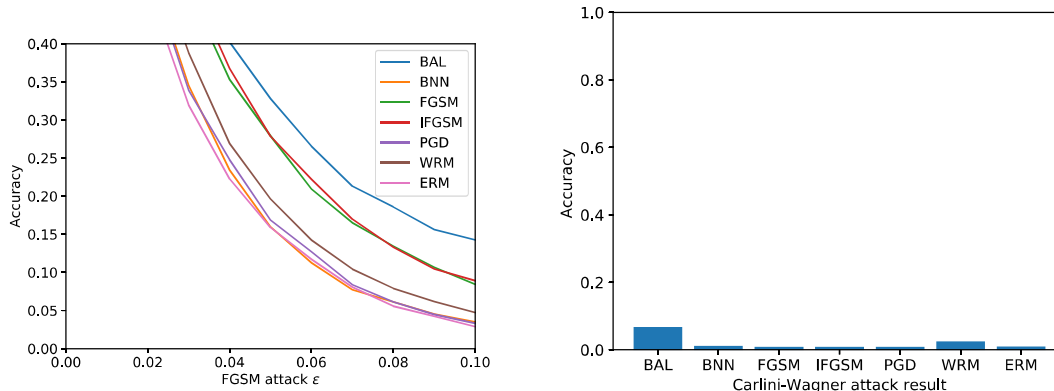

Figure 3: (a)Test accuracy for white-box attacks by FGSM method for traffic sign recognition (b)Test Accuracy for white-box attacks by Carlini-Wagner method for traffic sign recognition

# 6    Conclusion

We have proposed an adversarial training method, Bayesian Adversarial Learning, a full Bayesian treatment over adversarial training. Scalable sampling strategy is introduced to obtain the posterior distribution over the model parameters. We have shown that the proposed BAL has achieved the best performance in adversarial training for MNIST classification and a practical safety-critical problem, e.g., traffic sign recognition. In our experiments, the performance of BAL could always be better than the random guessing method even with the strong Carlini-Wagner attacks, which is important for safety-critical systems in real-world scenarios. For the future work, we will explore the variants of Bayesian adversarial learning on generative models, robust unsupervised and semi-supervised learning. Besides, we will also explore more efficient scalable samplers for Bayesian adversarial learning.

## Acknowledgments

Dr. Zhanxing Zhu is supported by National Natural Science Foundation of China (Grant No: 61806009) and Beijing Natural Science Foundation (Grant No: 4184090).

## Footnotes

[2]https://github.com/duchi-lab/certifiable-distributional-robustness

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
