[Supplementary Material · supplementary.pdf]

# A    Fashion-MNIST Classification

The Fashion-MNIST dataset consists of 60000 images for training, 10000 images for test with 10 categories including dress, coat, shirt, etc. We plot the results in Figure 4. Figure 4, we can observe that the BAL achieves better results in FGSM attack experiment but only get slightly worse results than WRM in Carlini-Wagner attack experiment. This is because training neural networks is much harder and might need much larger sample size for BAL in Fashion-MNIST than MNIST. [30]. We also observe that the numerical computation for Fashion-MNIST results is not very stable. We speculate that neural networks can over-fit the dataset easily as humans can only achieve 83.5% accuracy on the dataset and other methods except the BAL achieve much higher accuracy on the dataset.

Figure 4: **Left:** Test accuracy on white-box attacks generated by FGSM method for Fashion-MNIST classification; **Right:** Test accuracy on white-box attacks generated by Carlini-Wagner method for Fashion-MNIST classification