[Reviews · NeurIPS 2018]

Reviewer 1



This paper proposes a Bayesian model for adversarial learning problem. Empirical studies on Fashion-MINST and traffic sign recognition show that the proposed methods is slightly better than other adversarial learning baselines. Below I list my concerns about the paper: For modeling, 1. This paper ignore a highly relevant work ‘Bayesian GAN’ [1]. The non-cooperative game between ‘data generator’ and ‘learner’ established in this paper is almost the same as the vanilla GAN. And the fully Bayesian treatment introduced is exactly the same as in [1]. 2. The author mentioned the posterior inference over learner’s parameter can be achieved by Gibbs sampling on the two defined conditional posterior. The analysis of the underlying joint posterior distribution is missing. It is hard for readers to understand what actually the distribution over learner is achieved when the algorithm converges. 3. It also remains unclear that whether the iteration introduced in Eqn. (3) and (4) is guaranteed to converge. For inference, 1. In the consideration of the scalability, the author proposed to only update a mini-batch of training data points. However, no theoretical insight is provided to validate this stochastic approximation. Does this stochastic process on expectation match the iteration defined in Eqn. (3) and (4)? For experiments, 1. In section 5, the author suggests a sample size of 5 and claims that better performance can be achieved with a larger sample size. I recommend including results with different sample size to support the augment. Overall, I think this is an interesting paper exploring the functionals of Bayesian modeling in the context of adversarial learning. However, the novelty of proposed method is limited, and the relationship between the current work and [1] is totally ignored in the paper. I recommend weak rejection for this paper. [1] Bayesian GAN; Yunus Saatchi, Andrew Gordon Wilsons ----------------------------- Update: I have read the author feedback and I appreciate the authors’ reply on my concerns. I believe given the previous work ‘Bayesian GAN’, another way to frame the contribution would be adapting the techniques in Bayesian GAN for the problem of adversarial attacks. In terms of methodology, the only main difference is that Bayesian GAN generates samples conditioned on random noise while BAL generates samples conditioned on the input data (as mentioned in the first point of the author feedback). The other parts are very similar. The application itself is important and interesting. I am happy to see the Bayesian principle help design a good adversarial defense framework that achieves good empirical results. Based on the reasons above, I will slightly upgrade my score (from 5 to 6). If this paper is accepted, I hope my following comments about *target posterior distribution* can be addressed in the final version. Surely L88-L89 gives a formula of the parameter posterior. However I urge authors to articulate what is the conditional distribution p(\tilde{D} |D). Since in the previous text, only p(\tilde{D} |\theta, D) is defined. It is very important to have a clear idea about what is the target of the learning algorithm. Missing this critical part may confuse the reader. For the inference part, giving a verification of mini-batch sampling as the authors suggested in the rebuttal will also make this paper more self-contained.

Reviewer 2



This paper focusses on adversarial learning, that is learning of deep models that is robust to adversarial data, in a Bayesian framework. The usual approaches to adversarial learning consist is "point estimates", while the proposed approach averages, in a Bayesian sense, over a specified distribution on adversarial data-generating distribution. I find the idea of Bayesian averaging over the adversarial data-generating distribution interesting. However, the Practical Instantiation of Bayesian Adversarial Learning does not seem practical to me, and reproducing the obtained results not possible. Unless I am wrong, some parameters of Algorithm 1 are not specified, including \eta_1, \eta_2, \gamma, or C.

Reviewer 3



To the authors: I have read the rebuttal and it helped me gain better understanding of the paper. I maintain my rating on the paper in hope that it gets accepted. Adversarial machine learning has focused mainly on solving a minimax problem, with the inner part representing the worst loss that can be caused by an adversary that is constrained to perturb within a certain region or according to a certain distribution. A potential weakness, as is brought up as the motivation of this paper, is that when the adversary is beyond the pre-defined constraints, the model performance could deteriorate sharply. With this in mind, the authors proposed a Bayesian approach that impose a distribution over the adversarial data-generating distribution, and aim to learn the optimal posterior distribution of the learner given observed data, obtained by Gibbs sampling. While I think the idea is quite interesting and worth being considered for publication, I have a few questions and comments: 1. Why design the distribution over data distribution as (1) and (2)? 2. What is the consideration for the design in the paper where the learner uses a sampling strategy to determine its model parameter instead of performing optimization? 3. From the experiment result in the left-hand plot for both figures 1 and 3, it seems to me that BAL is not generating much better performance than the benchmarks. In fact, for small epsilon it is performing slightly worse. What is the potential cause behind this? 4. I think there are grammar errors in the paper. For example, Line 26 defense -> defend. The authors are advised to make a full check if the paper gets accepted.